# Liver Fibrosis Resolution: From Molecular Mechanisms to Therapeutic Opportunities

**DOI:** 10.3390/ijms24119671

**Published:** 2023-06-02

**Authors:** Qiying Pei, Qian Yi, Liling Tang

**Affiliations:** 1Key Laboratory of Biorheological Science and Technology, Ministry of Education, College of Bioengineering, Chongqing University, Chongqing 400044, China; 15334549373@163.com; 2Department of Physiology, School of Basic Medical Science, Southwest Medical University, Luzhou 646000, China

**Keywords:** liver fibrosis, fibrosis regression, hepatic stellate cells, therapeutic compounds

## Abstract

The liver is a critical system for metabolism in human beings, which plays an essential role in an abundance of physiological processes and is vulnerable to endogenous or exogenous injuries. After the damage to the liver, a type of aberrant wound healing response known as liver fibrosis may happen, which can result in an excessive accumulation of extracellular matrix (ECM) and then cause cirrhosis or hepatocellular carcinoma (HCC), seriously endangering human health and causing a great economic burden. However, few effective anti-fibrotic medications are clinically available to treat liver fibrosis. The most efficient approach to liver fibrosis prevention and treatment currently is to eliminate its causes, but this approach’s efficiency is too slow, or some causes cannot be fully eliminated, which causes liver fibrosis to worsen. In cases of advanced fibrosis, the only available treatment is liver transplantation. Therefore, new treatments or therapeutic agents need to be explored to stop the further development of early liver fibrosis or to reverse the fibrosis process to achieve liver fibrosis resolution. Understanding the mechanisms that lead to the development of liver fibrosis is necessary to find new therapeutic targets and drugs. The complex process of liver fibrosis is regulated by a variety of cells and cytokines, among which hepatic stellate cells (HSCs) are the essential cells, and their continued activation will lead to further progression of liver fibrosis. It has been found that inhibiting HSC activation, or inducing apoptosis, and inactivating activated hepatic stellate cells (aHSCs) can reverse fibrosis and thus achieve liver fibrosis regression. Hence, this review will concentrate on how HSCs become activated during liver fibrosis, including intercellular interactions and related signaling pathways, as well as targeting HSCs or liver fibrosis signaling pathways to achieve the resolution of liver fibrosis. Finally, new therapeutic compounds targeting liver fibrosis are summarized to provide more options for the therapy of liver fibrosis.

## 1. Introduction

The liver, the main organ in the body, is involved in many physiological activities, including the elimination of infections, xenobiotics, medicines, and other toxic chemicals; the metabolism of glucose, protein, and lipids; and immunological functions [1,2]. Despite the liver’s considerable capacity for regeneration [3], ongoing damage can bring about fibrosis, cirrhosis, and even hepatocellular carcinoma (HCC) [2]. Liver fibrosis, which results from an inappropriate response to liver injury’s wound-healing process, is a pathological condition [4,5], and the main pathological characteristics are the excessive accumulation of extracellular matrix (ECM) proteins and the loss of the normal structure of the liver tissue [6,7,8]. Additionally, various etiological factors can lead to liver fibrosis, such as viral infections, alcohol, metabolic diseases, biliary diseases, hepatic toxins [9], autoimmune disease [5], and prolonged use of some medications, including methotrexate, methyldopa, chlorpromazine, tolbutamide, etc. [8,10] (Figure 1).

Chronic liver illnesses, including viral hepatitis and fatty liver deteriorate in part due to liver fibrosis. Without adequate treatment, cirrhosis, which poses a major threat to human health, can arise in 75–80% of these conditions [11,12]. Patients with chronic liver disorders who progress to liver fibrosis and cirrhosis suffer high morbidity and mortality rates, placing a significant economic burden on society [8,13]. Liver fibrosis is typically a slowly progressing condition with no clinical symptoms. Although liver fibrosis originally happens as part of the liver repair process, if it is not controlled, it develops into a pathogenic condition [14]. With ongoing damage and ECM deposition, the liver is gradually hardening and stiffening over time, followed by a slow loss of function. Liver fibrosis may then progress into cirrhosis or even liver cancer, with a variety of other consequences involving portal hypertension and liver failure [15,16,17]. Currently, the only effective treatment for advanced liver disease is liver transplantation [15,18], but it has its limitations. In all etiologies of chronic liver illnesses, the main factor causing liver failure is liver fibrosis, and the degree of fibrosis is the most reliable indicator of mortality from liver-related causes [19,20,21,22]. Accordingly, discovering suitable strategies to prevent or reverse liver fibrosis is crucial to preventing further liver fibrosis deterioration.

The process of liver fibrosis is regulated by a variety of cells and cytokines [23]. When the liver is continuously damaged, the damaged cells release pro-inflammatory or pro-fibrotic factors, which in turn activate relevant signaling pathways and promote hepatic stellate cells (HSCs) activation, resulting in the production of large amounts of ECM, leading to further deterioration of liver fibrosis [16]. Most of the therapeutic drugs for liver fibrosis are still in clinical trials and research stages [23]. Hence, this review will concentrate on the activation of HSCs in liver fibrosis and the cellular and molecular pathways of liver fibrosis resolution. Furthermore, the new therapeutic compounds for liver fibrosis in recent years are summarized.

## 2. The Mechanisms of Liver Fibrosis

The continuous deposition of ECM [24], which harms the liver’s physiological structure [13] and impairs its capacity for normal activity [7], is the hallmark of liver fibrosis. Additionally, it is caused by highly active crosstalk among various cells, including hepatocellular parenchymal cells (hepatocytes), non-parenchymal cells (like Kupffer and endothelial cells), and immune cells [23,25,26]. Myofibroblasts (MFs) contribute to the production of ECM and are regarded as the primary effector cells in the development of fibrosis [4,27]. Although the source of activated MFs may vary depending on the different etiologies, HSCs can be considered their main origin [3]. Thus quiescent hepatic stellate cells (qHSCs) activation and subsequent conversion into MFs are essential occurrences in the pathogenesis of liver fibrosis [7,26,28]. Consequently, treating liver fibrosis requires limiting the activation of HSCs [7].

### 2.1. Hepatic Stellate Cells

HSCs are a unique group of pre-hepatic cells that are situated in the Disse space next to sinusoidal endothelial cells and hepatocytes [29]. HSCs in a healthy liver display a quiescent state, and their physiological roles involve fat storage and the metabolism of vitamin A [12]. The secretion of sufficient amounts of ECM proteins, including type III collagen, type IV collagen, and laminin, to maintain the homeostasis of ECM is another duty performed by the qHSCs [23].

In the process of developing liver fibrosis, HSCs are crucial [5,30]. MFs, which secrete matrix proteins and are the primary cause of liver fibrogenesis, are mostly derived from activated hepatic stellate cells (aHSCs) [6]. Several important cells and inflammatory mediators, such as inflammatory stimuli and fibrogenic cytokines like transforming growth factor-β (TGF-β), reactive oxygen species (ROS), etc., released by activated macrophages, platelets, and damaged hepatocytes, activate the qHSCs when the liver is injured [23,27]. Furthermore, non-coding RNAs (ncRNAs) can regulate the development of liver fibrosis by acting on HSCs. For example, circular RNAs (circRNAs) frequently function as miRNA sponges that control HSC activation and proliferation and hence participate in the development of liver fibrosis [31]. In addition, it has been found that exosomes, a type of extracellular vesicles (EVs), from various cell sources can also stimulate HSCs and play an important role in the process of liver fibrosis [32,33]. Exosomes can carry various cellular components, including nucleic acids, proteins, lipids, and other molecules involved in the transmission of information between cells [32,34]. It has been found that exosomal cellular communication network factor 2 (CCN2), derived from HSCs, contains connective tissue growth factors that promote the development of liver fibrosis and can amplify or attenuate fiber formation signals by binding to other exosomal components [35].

HSCs also perform a transdifferentiation process that changes them to a proliferative and contractile myofibroblast phenotype from a quiescent phenotype [13,36]. Additionally, in conditions of chronic injury, the continued activation of HSCs causes an imbalance between the deposition and dissolution of the ECM, which in turn causes liver fibrosis to proceed [13]. MFs may continue to be triggered by excessive cytokines, which result in the formation of massive ECM aggregates and excessive ECM deposition [23]. The abnormal deposition of ECM results in aberrant wound healing responses and then increases the formation and growth of liver fibrosis [27,36]. In conclusion, liver fibrosis is caused by, develops into, and can be reversed by HSCs. Moreover, a range of cells, including Kupffer cells (KCs), hepatocytes, and numerous signaling pathways, can regulate HSC activation [12].

### 2.2. The Intercellular Crosstalk of HSC Activation

For the preservation of healthy liver function and cell survival, intercellular interaction in the hepatic microenvironment is crucial [6]. Additionally, the progression of liver fibrosis, which is an intricate process of liver self-repair, is related to the highly active crosstalk among several cell types [25]. Although HSCs are thought to be the primary cell of action in liver fibrosis, other types of liver cells also contribute significantly to the condition. In fact, interaction with other hepatic cells, such as hepatocytes, liver sinusoidal endothelial cells (LSECs), and inflammatory cells, is necessary for HSC activation (Figure 1). Through the release of cytokines and other signaling molecules, these cells communicate with one another and either encourage or suppress the activation of HSCs [6].

#### 2.2.1. Hepatocytes

More than 80% of all liver cells are hepatocytes, the most significant parenchymal cells in the liver [6]. Hepatocytes perform a variety of tasks in physiological environments, including detoxification and the release of bile, proteins, and lipids [37]. Hepatocytes that have suffered injury generate substances that stimulate HSCs and inflammatory cells, which bring about the development of fibrosis [6].

One of the key initial stages in the pathophysiology of all liver illnesses and their etiologies is hepatocyte death. Damage-associated molecular patterns (DAMPs), which are intracellular substances that can be released by dying hepatocytes and act as warning signals to neighboring cells like KCs and HSCs, are essential for inflammation and the formation of fibrosis [13]. High-mobility group box-1 (HMGB1), a DNA-binding non-histone nuclear protein, is typically produced in eukaryotic cells and is one of the most researched DAMPs in liver disease [38]. Furthermore, it has been demonstrated that HMGB1 could directly activate HSCs by controlling HSC autophagy in liver fibrosis development models linked to the hepatitis B virus (HBV) [39]. Furthermore, hepatocytes can also enhance the activation of HSCs by secreting exosomes, thus promoting the progression of liver fibrosis [34]. Lipotoxic hepatocyte-derived exosomal miR-27a was found to promote the development of liver fibrosis in metabolically associated fatty liver disease (MAFLD) by enhancing HSC proliferation and activation and inhibiting aHSC autophagy [40].

#### 2.2.2. Liver Sinusoidal Endothelial Cells

LSECs mostly exhibit anti-inflammatory, anti-fibrotic, and regeneration-promoting actions in healthy liver tissue [41]. In addition, the interaction among HSCs, hepatocytes, and LSECs could maintain HSC quiescence and regenerate hepatocytes [6], which is essential for fibrosis development.

Studies have demonstrated that LSECs can maintain HSCs quiescence through a vascular endothelial growth factor (VEGF)-stimulated pathway [42]. When the liver is injured, the capillary production of LSECs both decreases the synthesis of vasodilators (such as NO and cyclooxygenase) and increases the synthesis of vasoconstrictors (such as endothelin 1 and thromboxane A2) [23]. Although changing the phenotype of LSECs, this imbalance also helps to activate HSCs, which in turn encourages inflammation and liver fibrosis [43,44]. Moreover, LSECs can release signaling molecules including TGF-β and platelet-derived growth factor (PDGF) or activate signaling pathways like Wnt/β-catenin, which in turn activate HSCs [23].

#### 2.2.3. Inflammatory Cells

Inflammation brought on by persistent liver injury can be triggered by the release of ROS and other signals from damaged cells, which in turn activate inflammatory cells like macrophages, lymphocytes, and others [45]. Additionally, liver fibrosis worsens due to the mutual activation of inflammatory cells and HSCs [13].

The predominant macrophage populations in the initial stages of the damage are pro-inflammatory [46]. Among them, hepatic macrophages (KCs), which are located within the lumen of the liver sinusoids [17], play crucial roles in liver fibrosis development [47]. When the liver is injured, the KCs become active and quickly secrete substances that promote inflammation and fibrosis, such as interleukin (IL)-1β, tumor necrosis factor-alpha (TNF-α), chemokine (C-C motif) ligand 2 (CCL2), and CCL5. This causes HSC activation and exacerbates liver fibrosis [6,13,48]. Additionally, the activation of KCs can boost HSCs’ nuclear factor kappa-B (NF-κB), which in turn encourages the release of pro-inflammatory cytokines [49].

### 2.3. The Molecular Mechanisms of HSC Activation

Multiple signaling pathways can modulate the activation of HSCs at the molecular level. Growth factors and ligand-receptor signaling pathways, profibrogenic response pathways, cell death signaling, immune-related signaling, metabolically regulated pathways, nuclear receptors, and epigenetic changes are the main signaling pathways involved in HSC activation [6]. The TGF-β signal transduction pathway, PDGF signal transduction pathway, Hippo signaling pathway, and ROS that act on the activation of HSCs are mostly summarized in this part (Figure 1).

#### 2.3.1. TGF-β Signaling Pathway

TGF-β, a crucial pro-fibrotic cytokine in the liver [50,51], includes TGF-β1, TGF-β2, and TGF-β3, among which TGF-β1 is the most potent of these proteins and the main regulator of chronic liver disease, increasing the illness’s progression from early liver damage to inflammation, fibrosis, and cirrhosis, even HCC [52,53]. Additionally, TGF-β1 is widely implicated in HSC activation and proliferation [13,23]. The production of laminin, α-smooth muscle actin (α-SMA), fibronectin, and fibrous collagen (mostly types I and III) is increased as a consequence of the most potent cytokine, TGF-β1. Additionally, its expression is relatively low in normal liver tissue, but when liver injury occurs, KCs and LSECs can produce enormous amounts of TGF-β1 to encourage the activation of HSCs, and then the aHSCs subsequently secrete TGF-β1 from themselves to act on themselves or other cells [16,51]. Both canonical (Smad) and non-canonical (non-Smad) pathways can be used by TGF-β to cause fibrosis [54].

A crucial mechanism in the induction of ECM formation during liver fibrosis is the TGF-β/Smads pathway [5,11,53]. Additionally, the two primary downstream mediators are Smad2 and Smad3 [55,56]. Intracellular signal transducers that specifically react to the modification of the TGF-β receptor make up the Smad superfamily [6]. According to their specific functions, the Smad protein family can be divided into three categories. Smad2 and Smad3 are Smads that are receptor-regulated (R-Smads) [13]. In TGF-β/Smad pathways, TGF-β will bind to the TGF-β receptor type II, which then phosphorylates TGF-β receptor type I, recruits and phosphorylates R-Smads (Smad2/3) [53], and then forms a complex with Smad4, which translocates into the nucleus and stimulates target gene transcription (like collagen-I) [53,57,58].

TGF-β may stimulate HSCs through non-Smad pathways as well as the Smad pathway, including mitogen-activated protein kinase (MAPK), extracellular regulated protein kinase (ERK), c-Jun N-terminal kinase (JNK), etc. For instance, kindlin-2 overexpression was related positively to the TGF-β signaling pathway, while activated TGF-β could promote kindlin-2 expression through the p38 and MAPK signaling pathways [59,60].

In summary, both Smad and non-Smad pathways have the potential to activate HSCs, which would then favor the formation and progression of liver fibrosis.

#### 2.3.2. PDGF Signaling Pathway

PDGF belongs to a group of growth factors whose physiological functions include controlling cell proliferation and migration and promoting the synthesis of vital components of the connective tissue matrix [23,61]. In normal conditions, platelets produce PDGF. Additionally, upon liver injury, KCs might facilitate the intrahepatic recruitment of platelets [13]. Additionally, aHSCs can also express PDGF, and these cells also express the PDGF receptor (PDGFR) on their surface. The activation of HSCs is then stimulated by PDGF using an autocrine pathway [62,63].

In addition, the phospholipase Cγ(PLCγ), the phosphatidylinositol 3-kinase (PI3K)/protein kinase B (Akt) pathway, and the signal transducer and activator of transcription (STAT) pathway can be triggered when PDGFR is stimulated. Once these downstream components control the expression levels of pro-fibrosis target genes, liver fibrosis is facilitated [10,64].

#### 2.3.3. Hippo Signaling Pathway

HSC activation can be regulated by the Hippo pathway [65]. Hippo is a signaling mechanism that modulates the size of cells and organs. Additionally, it is crucial for tissue homeostasis, carcinogenesis, and liver growth regulation during development [53,66]. Yes-associated protein 1 (YAP1), a Hippo pathway downstream effector, is essential for regulating HSC activation in response to chronic damage [53,66]. Yes-associated protein (Yap) is translocated into the nucleus upon hippo pathway kinase inactivation to encourage the transcription of genes like connective tissue growth factor (CTGF) and TGF-β [53]. Moreover, it has been demonstrated that Hippo pathway elements, including YAP1 and the protein kinases macrophage stimulation 1 (MST1) and MST2, are essential for the initial activation of HSCs [67].

#### 2.3.4. Reactive Oxygen Species

ROS can function as signal molecules in numerous signaling pathways, including transcriptional regulation, proliferation, and differentiation [68]. Excessive production of ROS can result in oxidative stress, which promotes a sustained inflammatory response as well as HSC activation and proliferation, which then leads to the appearance and growth of liver fibrosis [68]. ROS are produced as part of a cell’s regular metabolic activity [13]. Additionally, at low concentrations, ROS can function as secondary messengers to trigger various cellular reactions, which can alter cell growth, encourage apoptotic pathways, and aid in the removal of pathogens [13,67]. High ROS concentrations, however, have the potential to damage proteins and DNA, trigger hepatocyte necrosis and apoptosis, release mediators like TGF-β and TNF-α, and eventually lead to HSC activation [67]. In addition to encouraging the recruitment of circulating inflammatory cells into the liver, ROS can encourage KCs to release pro-fibrotic mediators, which in turn activate HSCs [10]. In summary, high levels of ROS are important for HSC activation.

## 3. Regression of Liver Fibrosis

Although previous studies have made great progress in understanding the pathogenesis and progression of fibrosis, the complicated pathogenesis of fibrosis has presented some challenges in developing anti-hepatic fibrosis medications. Additionally, earlier research supported the belief that fibrosis was an unreversible process, but a growing number of clinical investigations have demonstrated that fibrosis will have regression at least partially when the underlying cause is eliminated [3,17,69]. The liver has a great capacity for regeneration when compared to organs like the lung and kidney, and liver fibrosis has a greater possibility to regress and return to its normal structure than other tissues, which may make the therapy of liver fibrosis easier [3,14].

Recent research has demonstrated that liver fibrosis is typically reversible. Liver fibrosis is not a one-way, irreversible process, even when it is advanced. When the stimulation of liver injury is removed, the liver can activate repair mechanisms that inhibit MF activation and change immune cells, particularly macrophages, from being pro-inflammatory to a repair state [4,28]. Antiviral therapy, for instance, exhibited regression of fibrosis in patients with viral hepatitis-related liver fibrosis, indicating that the underlying cause of the fibrosis can be treated to reverse liver fibrosis [14]. It has been demonstrated that removing the primary cause of chronic inflammation might cause severe liver fibrosis brought on by chronic HBV and hepatitis C virus (HCV) infection to regress [13].

Consequently, it is necessary to develop more effective therapies to delay fibrosis progression or promote regression of liver fibrosis, which can help prevent the advancement of cirrhosis or cancer [12,69]. New treatment targets for liver fibrosis will be identified if the processes of liver fibrosis regression are understood. The main ways to achieve liver fibrosis regression [3] are to eliminate the factors causing liver damage [70], reduce the number of aHSCs [71], inhibit the signal pathways linked to liver fibrosis [72], and promote the degradation of the ECM [73] (Figure 1).

### 3.1. Reducing the Number of Activated HSCs

The modulation of cell levels, cytokines, and their molecular levels are all part of the intricate mechanism behind liver fibrosis [11]. Notably, stimulation of HSCs is a typical feature of liver fibrosis [11,68]. Eliminating the chronic liver injury causes liver fibrosis to regress along with a decrease or disappearance of aHSCs [74]. Moreover, clinical and experimental studies have both demonstrated that the disappearance of aHSCs by apoptosis [75], inactivation into a quiescent-like state [76,77], or senescence [78] may lead to the regression of liver fibrosis. HSCs may be the primary target of liver fibrosis regression due to their crucial role in the condition [79]. Consequently, reducing the number of aHSCs, including inhibiting their proliferation or activation [7,24], promoting HSC apoptosis or autophagy [11,68,80], and deactivating the activated HSCs [76,77] to reverse liver fibrosis, may be an anti-fibrosis strategy [81].

#### 3.1.1. Limitation of HSC Activation

The phenotype of qHSC can change from static to active when the liver is injured by inflammation or mechanical stimulation [82]. After that, aHSCs can express α-SMA and produce ECM components like collagen and laminin [83]. Moreover, aHSCs can release pro-inflammatory and pro-fibrotic molecules that aid in the development of liver fibrosis. The resulting aHSCs are a crucial part of the fibrosis response that is driven by numerous pathways and is one of the primary effector cells in liver fibrosis [58]. Hence, liver fibrosis can be prevented or reversed by preventing the proliferation or activation of HSCs.

Changes in the cell microenvironment, including soluble mediators, ECM composition, matrix hardness, and interactions with adjacent cells (e.g., damaged hepatocytes and immune cells), can significantly affect the activation of HSCs [3]. A metabolic enzyme called alcohol dehydrogenase (ADH) is mostly expressed in the liver, and ADHI is the classical liver ADH among these family members [84]. Research [85] has shown that the activity of ADHI in human fibrotic liver is significantly higher than that in normal liver. ADHI might encourage the proliferation, motility, adhesion, and invasion of HSC-T6 cells, a rat HSC line. Additionally, the liver fibrosis in mice caused by carbon tetrachloride (CCl_4_) can be attenuated by the ADH inhibitor 4-methylpyrazole (4-MP), which might be connected to the suppression of the activity of ADHI [86]. This suggests that inhibiting HSC activation can alleviate liver fibrosis. *Amydrium hainanense* (AH) is a plant drug, and its aqueous extract (AHWE) has an antifibrotic effect on mice with liver injury. The primary mechanism through which AHWE had this impact was by preventing HSC activation and proliferation [87]. The inhibition of circASPH, which is derived from the aspartate β-hydroxylase (ASPH) gene locus, can suppress the activation and proliferation of HSCs, while its upregulation promoted the occurrence of these events in a CCl_4_-induced mouse model of liver fibrosis [88]. CircDIDO1 (death-inducer obliterator 1) inhibits HSC activation by inhibiting the Akt pathway, and overexpression of circDIDO1 can induce cell cycle arrest and inhibit HSC proliferation and activation in the human HSC cell line LX-2 [89]. All of these provide a new approach to reversing liver fibrosis by inhibiting HSC proliferation or activation.

#### 3.1.2. Apoptosis of HSCs

HSCs’ aberrant activation and proliferation during the progression of liver fibrosis may accelerate the death of other healthy liver cells, leading to further deterioration of the disease [90,91]. A type of programmed cell death called apoptosis helps control the HSCs’ balance between growth and death during fibrosis [3]. Clinical trials showed that the number of apoptotic HSCs decreased with the fibrosis stage increasing, independent of HSC activation or proliferation [92]. Studies on animal models of liver injury induced by different methods have shown that reducing the number of HSCs by inducing apoptosis is directly related to the regression of fibrosis, suggesting that HSC apoptosis is also a major determinant for liver fibrosis regression [75,92]. Therefore, the reversal of liver fibrosis can be achieved by accelerating the death of aHSCs and reducing the total number of aHSCs [93,94].

The authors of [95] have shown that liver fibrosis treatment could rely on endoplasmic reticulum stress (ERS)-mediated apoptosis. Additionally, the ERS inducer tunicamycin (TM) can cause the apoptosis of HSCs through the calpain-2/Ca^2+^-dependent ERS pathway in the HSC line HSC-T6 [95]. An antiviral medication named tenofovir disoproxil fumarate (TDF) is frequently used to treat chronic hepatitis B (CHB). Additionally, recent clinical studies have shown that TDF treatment can not only suppress the virus but also make liver fibrosis regress. In a mouse model of liver fibrosis, TDF leads to aHSC apoptosis by downregulating the PI3K/Akt/mammalian target of rapamycin (mTOR) signaling pathway and ultimately improving liver fibrosis [96]. In the hunt for medications to treat or reverse liver fibrosis, it is crucial to investigate the process causing HSC apoptosis.

#### 3.1.3. Inactivation of HSCs

Studies have shown that over half of mouse and human HSCs return to a deactivated or inactive state during the reversal of liver fibrosis and avoid apoptosis or cell senescence [76,97]. Consequently, targeting the deactivation of aHSCs has become a novel and effective method for liver fibrosis resolution [76,97].

The inactivation of the zinc finger transcription factor GATA4 in HSCs has been demonstrated to result in aberrant ECM component formation and HSC activation in the embryonic liver [97]. Research has discovered that reactivation of GATA4 could change aHSCs into a quiescent phenotype and then reverse liver fibrosis [83]. Thus, GATA4 may be an appropriate target for the resolution of liver fibrosis. Peroxisome proliferator-activated receptor-γ (PPAR-γ), which is a major regulator of the adipogenesis process, has been found to associate with the activation of HSCs [77]. PPAR-γ levels are noticeably reduced during HSC activation [98], and overexpression of PPAR-γ in aHSCs induces a transition to a quieter phenotype [99,100]. Transcription factor 21 (Tcf21) has been found to be a deactivation factor of fibrogenic HSCs [101]. In vitro and in vivo tests, overexpression of Tcf21 in aHSCs not only suppressed fibrosis-related gene expression but also restored the cells to a quiescent phenotype, at least partially. These phenotypic alterations were accompanied by the regression of liver fibrosis and improved liver function [101]. Thus, Tcf21 may be an option for therapy for the reversal of liver fibrosis.

Our understanding of the underlying molecular processes that lead to aHSC inactivation remains limited. Therefore, the molecular mechanism of aHSC inactivation still needs to be investigated to discover additional candidates for reversing liver fibrosis.

### 3.2. Inhibiting the Liver Fibrosis-Related Signal Pathways

Liver fibrosis, including the activation of HSCs, involves multiple molecular mechanisms and signaling pathways, and disruption of these pathways helps to reverse the liver fibrosis process. Consequently, the regression of liver fibrosis can be achieved by interfering with these signaling pathways [79,102]. Many signals are crucial to the progression of liver fibrosis, including TGF-β [103], Wnt/β-catenin [104], Notch [105], Hedgehog [106], Hippo [93,107], and inflammasome signaling pathways [108]. 

The most potent fibrogenic cytokine, TGF-β, has an influence on all stages of disease progression, from primary liver injury to fibrosis and cirrhosis [14,109]. During liver fibrosis, TGF-β is upregulated, and its main role is to activate HSCs. It can also enhance the synthesis of tissue inhibitors of matrix metalloproteinases (TIMPs) and directly promote the production of interstitial fibrillar collagens [110]. Therefore, liver fibrosis can be efficiently prevented by inhibiting TGF-β or disrupting its downstream signaling pathway [51]. Over time, various tactics to inhibit TGF-signaling have been investigated, including targeting TGF-β isomers or suppressing TGF-β receptor activation [110]. In addition, many drugs have been discovered in recent years to regulate fibrosis progression by affecting the TGF-β signaling pathway. For example, Jageum–Jung (JGJ), a possible medication for treating liver fibrosis, can inhibit HSC activation by regulating the TGF-β1/Smad signaling pathway [109]. GNS561 is a novel oral anticancer drug with high hepatotropic properties, and it prevents the activation of HSCs and decreases the deposition of ECM by interfering with TGF-β1 maturation and TGF-β1/Smad signaling [16]. Due to its several crucial functions in immunological control, wound healing, cell differentiation, and proliferation regulation, targeting TGF-β is likely to have unintended harmful effects [14]. Therefore, there is a need to target specific steps of TGF-β activation in a more localized manner to lessen general toxicity. In addition, more targets need to be found to provide more options for liver fibrosis reversal.

Liver fibrosis can also be alleviated by targeting the Notch signaling pathway, so the Notch signaling pathway may be one of the targets for reversing hepatic fibrosis [6]. For example, costunolide (COS), which is isolated from *Saussurea lappa*, is involved in the WW domain-containing protein 2 (WWP2)-mediated notch homologue protein 3 (Notch3) degradation through ubiquitin-dependent lysosome pathways and subsequent restriction of the Notch3-hairy/enhancer of split-1 (HES1) pathway, thereby alleviating liver fibrosis [111]. In addition, PDGF and its receptors, ROS, YAP/transcriptional co-activator with PDZ-binding motif (TAZ), and other signaling pathways can be used as therapeutic targets for hepatic fibrosis regression.

In addition to the above pathways, the reversal of liver fibrosis can also be achieved through the direct promotion of ECM degradation, immune cells and their signaling pathways, stem cell transplantation, and other methods. However, more clinical trials are needed to reverse liver fibrosis.

## 4. The Therapeutic Compounds of Liver Fibrosis

Liver fibrosis directly damages liver function, is an essential stage before cirrhosis and is closely related to liver cancer. Numerous cell types, signaling pathways, and chemicals can be thought of as potential therapeutic targets because they all have a significant impact on the emergence, development, and resolution of liver fibrosis [8]. However, there is currently no clinically approved effective anti-fibrosis therapy for slowing or reversing liver fibrosis [7,112]. Treatment targeting the cause is thought to be the most effective method to prevent and treat liver fibrosis [57]. Though liver fibrosis may be slowed or reversed by eliminating the pathogen, it often happens too slowly or too little to avoid life-threatening complications. It is feasible to treat liver fibrosis with medicinal intervention, as some anti-fibrosis medications have demonstrated a potential function in inhibiting the advancement of anti-fibrosis in animal studies [12]. Certain compounds currently in clinical development have limited clinical efficacy, so the medical need to develop more effective and safer anti-fibrosis drugs remains high [102]. Therefore, we summarize the compounds that have been found to have therapeutic effects on hepatic fibrosis in recent years (Figure 2).

### 4.1. Natural Compounds

Natural compounds have gained prominence globally in recent years as one of the most significant sources of anti-fibrosis medications due to their affordability and accessibility [8,57]. Several studies have reported that natural products, like terpenoids, phenols, alkaloids, etc., as well as crude extracts or isolated molecules from plants, can alleviate various fibrosis diseases in vitro and in vivo [57]. Here is a summary of natural compounds that have been suggested to reduce liver fibrosis and their primary mechanisms of action (Table 1).

A variety of cancers, inflammation, and viral infections can be frequently managed with herbal plants. 18β-Glycyrrhetinic acid (18β-GA), a triterpenoid aglycone, is one of the chemical components of the natural product Glycyrrhizic acid [19,113,114]. Additionally, in mice with liver fibrosis brought on by bile duct ligation (BDL), 18β-GA can reduce liver fibrosis by causing ROS-mediated apoptosis in aHSCs [19]. In traditional Chinese medicine, Gan Huang Cao is a Miao plant used for medicinal and edible purposes [115]. Additionally, 88 substances, like flavonoids, organic acids, and coumarins, were extracted and identified from it by the researchers. The primary component of flavonoids, kaempferol (KA), has a variety of anti-inflammatory, antioxidant, and anti-apoptotic properties [115,116]. Studies have demonstrated that kaempferol could effectively attenuate the development of liver fibrosis [117,118]. The findings showed that kaempferol could inhibit HSC activation and suppress HSC collagen synthesis via downregulating the levels of phosphorylated Smad2 and Smad3, as well as the TGF-β1/Smads pathway [117]. Kaempferol can also inhibit HSC activation through the miR-26b-5p/Jag1 axis and the Notch pathway [118]. Therefore, kaempferol may be a novel natural compound to treat liver fibrosis. Ferulic acid (FA) is a polyphenolic compound that is abundant in grains, vegetables, and plants. FA is also among the primary active ingredients in various Chinese herbs like *Angelica sinensis*. FA exhibits a broad spectrum of biological functions, including antioxidant, enhancement of immune function, anti-inflammatory, and hepatoprotective properties [25,119]. FA has been found to alleviate liver fibrosis in rats by CCl_4_-induced liver fibrosis. In terms of molecular mechanisms, FA can reduce the activation of HSCs that is brought on by TGF-β1 through the inhibition of Smad2/3 phosphorylation and subsequent Smad4 signal transduction. Blocking these pathways could contribute to reversing the liver fibrosis process [120].

**Table 1 ijms-24-09671-t001:** Anti-fibrotic natural compounds.

Compounds	Animal Model	Targets/Pathways/Mechanisms	Reference
18beta-glycyrrhetinic acid (18β-GA)	BDL-induced liver fibrosis in mice	Targeting PRDX1/2	[19]
		Resulting in the accumulation of cellular ROS	
		Inducing apoptosis in activated HSCs	
Kaempferol (KA)	CCl_4_-induced liver fibrosis in female C57BL/6 mice	Inhibiting Notch pathway via miR-26b-5p/Jag1	[116,117,118]
		Downregulation of TGF-β1/Smad2/3 signaling	
		Inhibiting HSC activation	
Swertia purpurascens Wall extract (SPE)	CCl_4_-induced liver fibrosis in male Wistar rats	Inhibition of TGF-β/Smad/NF-κB signaling	[9]
		Inhibiting oxidative stress and inflammation	
Demethylzeylasteral (T-96)	CCl_4_-induced liver fibrosis in male C57BL/6J mice	Suppressing AGAP2-mediated FAK/Akt signaling	[121]
		Inhibition of downstream TGF-β1 induced fibrosis-related gene expression	
Ferulic acid (FA)	CCl_4_-induced fibrotic models in male C57BL/6J mice	Inhibiting hepatic oxidative stress, macrophage activation, and HSC activation	[25,120]
	CCl_4_-induced liver fibrosis in male Wistar rats	Through PTP1B and AMPK signaling pathways.	
		Inhibition of the TGF-β1/Smad pathway	
Methoxyeugenol	CCl_4_-induced liver fibrosis in male BALB/c mice	Modulating the activated phenotype of HSCs by activating PPAR-γ	[57]
		A protective effect against oxidative stress damage in hepatocytes	
Benzoquinone derivatives	TAA-induced liver fibrosis in male Balb/C mice	Ameliorating oxidative stress and inflammation	[68]
		Inducing apoptosis in activated LX-2 cells	
		Attenuating liver fibrosis in TAA-induced mice	
Carnosol (CS)	CCl_4_-induced liver fibrosis in male SD rats	Activation of SIRT1/EZH2	[28]
		Inhibiting HSC activation and reversing EMT	
Carnosic acid (CA)	BDL-induced liver fibrosis in male SD rats	Modulating the miR-29b-3p/HMGB1/TLR4 signaling pathway	[7]
		Attenuating BDL-induced liver fibrosis	
Stachydrine (STA)	CCl_4_-induced liver fibrosis in male SD rats	Inhibiting inflammation and oxidative stress	[122]
		Upregulating the ratio of MMPs/TIMPs to promote the degradation of ECM	
Catalpol	CCl_4_-induced liver fibrosis in male SD rats	Activating autophagy to exert anti-inflammatory effects	[8]
		Inhibiting the activation of HSCs	
Mogroside IVE (MGIVE)	CCl_4_-induced liver fibrosis in male C57BL/6 mice	Inhibiting the TLR4 signaling pathway	[112]
		Reducing inflammatory responses	
		Inhibiting TGF-β1 induced HSC activation	
Morin	DEN-induced liver fibrosis in male Wistar rats	Activating HIPPO signaling	[53]
		Downregulating TGF-β signaling	
		Attenuating fibrillar collagen deposition	
		Preventing HSC activation	
S-allyl cysteine (SAC)	CCl_4_-induced liver fibrosis in male Wistar rats	Inhibiting the expression of TGF-β	[123]
		A decrease in the profibrogenic cytokine, TGF-β	
		Correcting cytokine abnormality	
SMND-309	CCl_4_-induced liver fibrosis in male SD rats	Suppressing the expression of CTGF	[51]
		Scavenging lipid peroxidation products	
		Increasing endogenous anti-oxidation enzyme activity	
Quercetin (QE)	CCl_4_-induced liver fibrosis in male C57 mice	Suppressing the TGF-β1/Smads signaling pathway	[124]
	BDL-induced liver fibrosis in male C57 mice	Activating the PI3K/Akt signaling pathway to inhibit autophagy	
		Attenuating HSC activation	
Tormentic acid (TA)	CCl_4_-induced liver fibrosis in male SD rats	Suppressing HSC activation	[125]
		Suppressing the PI3K/Akt/mTOR signaling pathway	
		Inhibiting the NF-κB signaling pathway	
Gomisin D	CCl_4_-induced liver fibrosis in male Balb/C mice	Inhibiting HSC proliferation and activation	[82]
		Promoting HSC apoptosis	
		Regulating the PDGF-BB/PDGFRβ signaling pathway	
Costunolide (COS)	CCl_4_-induced liver fibrosis in male Balb/C mice	Regulating the Notch3–HES1 pathway	[111]
	BDL-induced liver fibrosis in male SD rats	Disturbing the PPM1G/WWP2 complex	
		Blocking the inhibitory effect of PPM1G on WWP2	
5-Methoxytryptophan (5-MTP)	CCl_4_-induced liver fibrosis in male SD rats	Regulation of the FOXO3a/miR-21/ATG5 pathway	[126]
		Inducing autophagy and suppressing HSCs’ activation	
Physalin B	CCl_4_-induced liver fibrosis in male C57BL/6J mice	Disrupting LAP2α/HDAC1 complexes	[127]
	BDL-induced liver fibrosis in male C57BL/6J mice	Inhibiting HDAC1-mediated GLI1 deacetylation	
		Inhibiting HSC activation	
Ginsenoside Rg1 (G-Rg1)	CCl_4_-induced liver fibrosis male C57BL/6J mice	Attenuating IDO1-mediated inhibition of maturation of DCs	[128]
		Inhibiting the proliferation of HSCs	
Germarcone (GER)	MCS/MCD diet induction in male C57BL/6 mice	Reducing the release of ROS	[129]
		Regulating TGF-β/Smad and apoptosis pathways	
		Inhibiting the activation and survival of HSCs	
Sweroside	CCl_4_-induced liver fibrosis in male C57BL/6 mice	Regulation of FXR-miR-29a signaling pathway	[130]
		Inhibition of HSC proliferation	
Artesunate	CCl_4_-induced liver fibrosis in male ICR mice	Activating HSC ferroptosis	[91]
		Inhibiting HSC activation	
Linderalactone (LIN)	CCl_4_-induced liver fibrosis in C57BL/6 mice	Inhibiting TGF-β/Smad signaling	[131]
		Suppressing HSC activation	

In recent decades, Chinese herbal formulas, including more than two kinds of Traditional Chinese Medicine (TCM), have been extensively utilized to treat diseases of the liver due to their synergistic therapeutic effect. Hence, Chinese herbal formulations have been researched for their anti-liver fibrosis effects [7,69,132]. Fuzheng Huayu (FZHY) formula, a formulation from TCM, is often utilized to treat chronic liver diseases in China [133,134]. Additionally, it is composed of *Salvia miltiorrhiza*, *Prunus davidiana*, *Schisandra chinensis*, *Pinus massoniana*, and *Gynostemma pentaphyllum*, etc. Additionally, FZHY has been demonstrated to improve liver fibrosis by targeting more than one molecule, like the TGF-β/Smad signaling pathways and TNF-α-induced apoptosis of hepatocytes [135,136]. A new TCM formula, named JY5, was obtained by the analysis of the FZHY’s primary active components, including salvianolic acid B, schisantherin A, and amygdalin [137]. Additionally, this study also demonstrated that JY5 blocked HSC activation by suppressing the Notch signaling pathway to alleviate liver fibrosis in animal models of liver fibrosis. Table 2 summarizes more Chinese herbal formulations and their main anti-fibrosis mechanisms.

### 4.2. Synthetic Compounds

To meet the needs of liver fibrosis treatment, a variety of highly effective compounds or improved derivatives can also be synthesized by chemical means, such as small molecular compounds and pyrazole derivatives.

PI3K is an essential signaling molecule that regulates numerous cellular functions [141]. When stimulated, PI3K is activated and phosphorylated to promote Akt phosphorylation, which can regulate multiple apoptosis-related signal transduction pathways [125]. Moreover, the PI3K/Akt pathway is also tightly linked to the incidence and development of liver fibrosis [139]. In CCl_4_-induced liver fibrosis mice, PI3K/Akt signaling inhibition can ameliorate liver fibrosis [139]. HS-173 [142] is a PI3K inhibitor. Additionally, the study has demonstrated that it could block the PI3K/Akt signaling pathway to induce HSC apoptosis and inhibit proliferation, which can attenuate the progression of liver fibrosis [142]. Consequently, HS-173 may be a useful medicinal compound for treating liver fibrosis. The derivatives of barbituric acids, a class of aromatic hydrocarbons, are crucial in biology and medicine [143]. Antioxidant, anti-inflammatory, and other biological activities are a few of the biological effects of barbituric acids and their derivatives [143,144]. Novel barbituric acid (BA) derivatives have been found to be able to prevent rats from developing non-alcoholic fatty liver disease (NAFLD) [145]. BA derivatives may therefore be a potential medicinal compound for treating liver fibrosis. BA-5 was screened for six distinct barbiturate derivatives’ antifibrotic effects on activating HSCs induced by TGF-β1 [102]. Additionally, further research revealed that BA-5 could block the TGF-β1 signaling pathways and the NF-κB signaling pathways induced by lipopolysaccharide (LPS) to inhibit HSC activation and liver fibrosis. Moreover, BA-5 can also inhibit macrophage recruitment and activation [102]. Altogether, BA-5 may be an efficient synthetic compound for liver fibrosis due to its dual therapeutic potential. Table 3 summarizes more synthetic compounds and their main anti-fibrosis mechanisms.

### 4.3. Drug Delivery System

Liver fibrosis, a serious disease, is a major threat to human health. With continuous, in-depth research, numerous anti-fibrosis medications have been found and show potent anti-fibrosis actions in vitro. However, there is no effective drug to treat liver fibrosis in the clinic. The reason is that most therapeutic medicines lack specificity, have limited solubility, and have poor liver accumulation effects. Accordingly, therapeutic drugs cannot reach an effective therapeutic concentration in vivo and are easily absorbed by non-target organ cells. For example, silymarin and curcumin are very effective against liver fibrosis. However, their application is restricted due to their low bioavailability, poor water solubility, lack of specific targeting, and additional liver toxicity [15,154]. To improve the efficacy of drug therapy, drug delivery systems, especially nanoparticles (NPs), have attracted extensive attention.

Exosomes are nanosized extracellular vesicles, usually between 30 and 150 nm in size, and some exosomes derived from stem cells can reduce liver fibrosis by carrying components that regulate HSC activation [155,156]. Exosomes can cross biological barriers to transfer nucleic acids, proteins, and other bioactive molecules [157]. Therefore, exosomes can also be used as delivery carriers for the treatment of hepatic fibrosis [34]. Osteopontin (OPN) is a matrix-bound protein that plays a significant role in liver fibrosis [158]. The exosomes carry small interfering RNA (siRNA) targeting OPN (Exo^siRNA-OPN^) and can deliver siRNA safely and effectively into the liver to reduce OPN expression. Exo^siRNA−OPN^ was found to inhibit TGF-β1 signaling by reducing HMGB1, then to attenuate the progression of liver fibrosis in a CCl_4_-induced mouse model of liver fibrosis [157].

With the development of nanotechnology, NPs provide new therapeutic opportunities for drug delivery to treat liver fibrosis. The nanomedicine delivery system can be employed to specifically target therapeutic agents to key cell types to alleviate or reverse liver fibrosis and achieve combination therapy with little systemic toxicity [1,159]. NPs have advantages such as adjustable dimensions and shapes, various compositions, and changeable surface characteristics, which can control drug release, improve drug pharmacokinetics, reduce adverse reactions, and promote targeted drug delivery and combination treatment of liver fibrosis, so as to solve the problem that the anti-fibrosis effect of traditional drugs is limited in vivo [15,159,160]. Nano-drug delivery systems are frequently divided into inorganic and organic NPs based on their natural properties. Metal ion NPs, such as Ag NPs and inorganic oxide NPs, are instances of inorganic NPs [1]. However, the biodegradability of inorganic NPs in vivo is poor, and the leftover NPs that accumulate there can harm the liver. Additionally, organic polymer materials like polyethylene glycol (PEG), polylactic acid (PLA), etc. can be used to synthesize organic NPs. The majority of organic NPs exhibit strong in vivo biodegradability, low toxicity, and few side effects [1]. Organic NPs, including lipid nanoparticles (LNPs), polymer NPs, and protein NPs, have been widely studied in treating liver fibrosis. These NPs can target the delivery of therapeutic drugs to the key cells of liver fibrosis, especially HSCs, thereby reducing the adverse reactions of drugs and improving the therapeutic effect of drugs in vivo.

TNF-stimulated gene 6 (TSG-6) was identified as a pivotal antifibrosis cytokine of mesenchymal stem cells that showed strong antifibrotic activity. Studies have shown specific liver targeting and high TSG-6 loading efficacy of calcium phosphate (CaP) @ bovine serum albumin (BSA) NPs. The CCl_4_-induced fibrosis in mice therapeutic study further demonstrated the improved therapeutic effects of TSG-6-loaded CaP@BSA [161]. LNPs include stable nucleic acid LNPs (SNALPs), phospholipids, steroid-based drug coupling liposomes, nanostructured lipid carriers (NLCs), etc. [162]. For example, the main peptide component found in honeybee venom, melittin, has been shown to be useful in lowering aHSC populations in fibrotic liver tissues [163]. However, the systemic toxicity caused by its nonspecific action limits its clinical application in fibrosis. To reduce the nonspecific effects of melittin, targeted drug delivery systems can be used to enable it to work in just fibrotic tissue. Fibroblast activation protein (FAP) is specifically overexpressed on the surfaces of aHSCs in the fibrotic liver [164]. This study [163] designed a delivery system to release melittin into the fibrotic liver surroundings by cleaving promelittin exclusively in liver areas that express FAP. Additionally, promelittin was attached to the PEGylated liposomes’ surfaces to improve liver transport and bloodstream stability. The results showed that promelittin-modified liposomes (PRL) are selectively activated in the fibrotic liver, and the FAP-specific liberation of melittin can decrease the number of aberrant HSCs and reduce the excessive production of collagen fibers to alleviate fibrosis. Moreover, PRL may be designed to increase antifibrotic efficacy by encasing additional antifibrotic chemical compounds in their core. In addition, the vitamin A-coupled liposomal Rho/Rho-associated protein kinase (ROCK) inhibitor may reduce liver fibrosis without causing systemic side effects while inhibiting HSC activation by targeting HSCs for drug delivery [165]. The SB431542-loaded liposomes showed better efficacy for treating liver fibrosis by inhibiting TGF-β signaling. Loading SB431542 into liposomes solves the limitations of poor water solubility and low bioavailability of this small-molecule drug [166].

However, the clinical transformation of nanomedicine for anti-fibrosis therapy is still complex in terms of specific targeting, safety, etc. More understanding of potential targets for hepatic fibrosis is therefore needed to identify promising anti-fibrosis drug candidates. Hence, the clinical application of nano-drug delivery systems must overcome many challenges.

## 5. Conclusions

Liver fibrosis is the organism’s attempt to repair chronic liver damage brought on by various pathogenic agents. With the in-depth study of liver damage and fibrosis mechanisms in the past few decades, it has been discovered that liver fibrosis is in a reversible stage and the liver has a strong regenerative ability. If timely treatment is carried out during this period, liver fibrosis reversal can be achieved and the liver can recover to a mostly healthy state. Therefore, it is important to diagnose and treat liver fibrosis early and promptly for the prevention of cirrhosis and liver cancer. However, since various cells in the body, including immune cells and signaling pathways, play dual roles in different liver microenvironments, it is crucial to further investigate the molecular mechanism of hepatic fibrosis regression to find more drug targets and efficient anti-fibrosis drugs, which are crucial for the effective treatment of hepatic fibrosis.

## Figures and Tables

**Figure 1 ijms-24-09671-f001:**
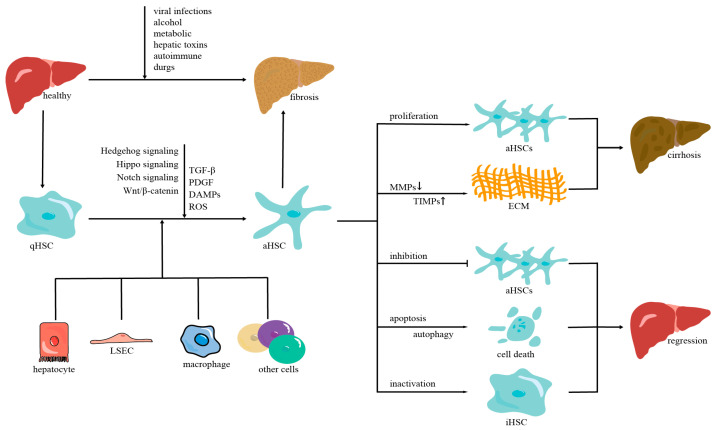
Hepatic stellate cells in the progression and regression of liver fibrosis. Quiescent hepatic stellate cells are activated when the liver is damaged. Many cells, including hepatocytes, liver sinusoidal endothelial cells, macrophages, TGF-β, and PDGF signal pathways, are involved in this process, which leads to the proliferation of aHSCs, and excessive secretion of ECM proteins, and eventually cirrhosis. Inhibition of HSC proliferation, induction of HSC death, or inactivation of HSCs can reverse hepatic fibrosis. qHSCs: quiescent hepatic stellate cells. TGF-β: transforming growth factor-β. PDGF: platelet-derived growth factor. DAMPs: damage-associated molecular patterns. ROS: reactive oxygen species. aHSCs: activated hepatic stellate cells. LSEC: liver sinusoidal endothelial cell. MMPs: matrix metalloproteinases. TIMPs: tissue inhibitors of matrix metalloproteinases. ECM: extracellular matrix. iHSC: inactivated hepatic stellate cell.

**Figure 2 ijms-24-09671-f002:**
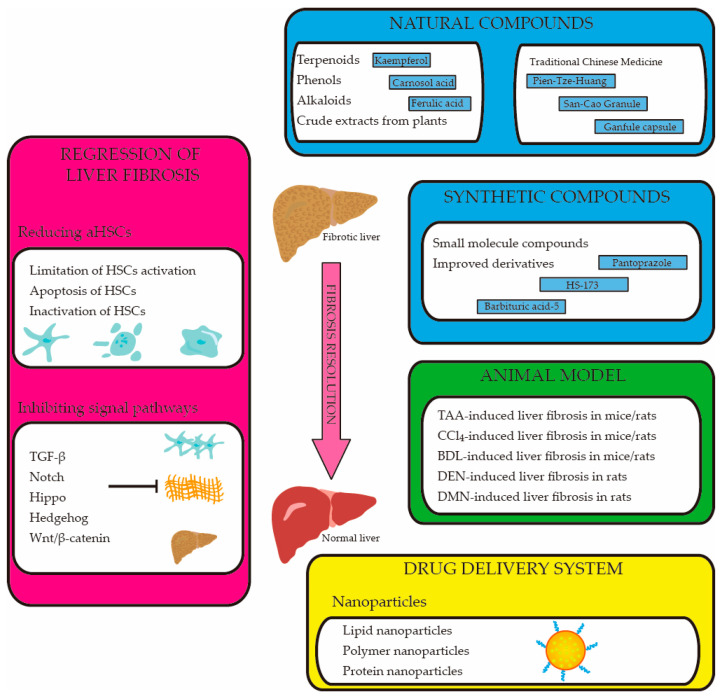
Therapeutic compounds for liver fibrosis resolution. Inhibition of HSC proliferation, induction of HSC death, or inactivation of HSCs and inhibition of liver fibrosis-related signal pathways can reverse liver fibrosis. Various compounds, including natural and synthetic compounds, have been found to have therapeutic effects in various animal models of liver fibrosis. Meanwhile, in order to improve the utilization rate of therapeutic drugs, a variety of nano-drug delivery systems are also used in the treatment of liver fibrosis. aHSC: activated hepatic stellate cell. HSC: hepatic stellate cell. TGF-β: transforming growth factor-β. TAA: thioacetamide. CCl_4_: carbon tetrachloride. BDL: bile duct ligation. DEN: diethylnitrosamine. DMN: dimethylnitrosamine.

**Table 2 ijms-24-09671-t002:** Anti-fibrotic Chinese herbal formulations.

Compounds	Animal Model	Targets/Pathways/Mechanisms	Reference
Pien-Tze-Huang (PZH)	CCl_4_-induced liver fibrosis in male SD rats	Autophagy and TGF-β1/Smad2 signaling pathways	[80]
		Inhibiting HSC activation and diminishing fibrogenesis	
Compound Kushen injection (CKI)	CCl_4_-induced chronic liver fibrosis in C57BL/6 mice	Inhibiting HSC activation	[138]
	MCD diet-induced NASH model in C57BL/6 mice	Rebalancing TGF-β/Smad7 signaling	
	CCl_4_-induced HCC models in C57BL/6 mice		
JY5	CCl_4_-induced liver fibrosis in male Wistar or SD rats	Inhibition of the Notch signaling pathway	[137]
	BDL-induced liver fibrosis in male Wistar or SD rats	Inhibiting the activation of HSCs	
	CCl_4_-induced liver fibrosis in male C57/BL6 mice		
Pokeweed antiviral protein (PAP)	CCl_4_-induced liver fibrosis in male C57/BL6 mice	Inhibition of glycolysis through regulation of the Wnt/JNK pathway	[24]
		Suppressing HSC activation	
San-Cao Granule (SCG)	TAA-induced liver fibrosis in SD rats	Inhibiting the TGF-β1/Smad pathway	[132]
CGA	DMN-induced liver fibrosis in male Wistar rats	Downregulation of MMP2/9 activities and TIMP1/2 protein expression	[136]
		Inhibition of the TGF-β1/Smad signaling pathway	
		Inhibiting the activation of HSCs	
Dahuang Zhechong pill (DHZCP)	CCl_4_-induced liver fibrosis in male SD rats	Inactivating the PI3K/Akt pathway	[139]
		Suppressing HSC proliferation	
Jageum–Jung (JGJ)	CCl_4_-induced liver fibrosis in male C57/BL6 mice	Inhibiting the activation and translocation of STAT-1 and NF-κB	[109]
		Regulating TGF-β1/Smad signaling.	
		Inhibiting HSC activation	
Ganfule capsule (GFL)	BDL-induced liver fibrosis in male C57/BL6 mice	Inhibiting glutamine metabolism	[140]
		Restricting the activation of the NF-κB pathway	

**Table 3 ijms-24-09671-t003:** Anti-fibrotic synthetic compounds.

Compounds	Animal Model	Targets/Pathways/Mechanisms	Reference
N-n-Butyl haloperidol iodide (F2)	TAA-induced liver fibrosis in C57BL/6J mice	Reducing responsiveness of HSCs to TGF-β1	[146]
	CCl_4_-induced liver fibrosis in C57BL/6J mice	Inhibiting activated HSCs	
Butaselen (BS)	CCl_4_-induced liver fibrosis in male BALB/c mice	Inducing the apoptosis of activated HSCs	[147]
		Inhibiting the production of α-SMA and collagens by HSCs	
		Downregulating TGF-β1 expression and blocking the TGF-β1/Smads pathway	
barbituric acid (BA)-5	CCl_4_-induced liver fibrosis in male C57BL/6 mice	Blocking both the TGF-β1 and LPS-induced NF-κB signaling pathways	[102]
		Inhibiting HSC activation and liver fibrosis	
		Inhibiting macrophages recruitment and activation	
JD5037	CCl_4_-induced liver fibrosis in male mice	Blocking the CB1 receptor/β-arrestin1/Akt signaling pathway	[79]
	BDL-induced liver fibrosis in male mice	Blocking the activation of HSCs	
GNS561	DEN-induced liver fibrosis in male rats	Disrupting TGF-β1 maturation and TGF-β1/Smad and MAPK signaling	[16]
		Inducing the apoptosis of HSCs	
		Preventing HSC activation and decreasing ECM deposition	
Cultured bear bile powder (CBBP)	DMN-induced liver fibrosis in wild-type Wistar rats	Elevating the expression of PPAR-α and PPAR-γ	[69]
		Improving β-FAO and inhibiting inflammation	
Vatalanib	CCl_4_-induced liver fibrosis in male BALB/c mice	Reducing liver inflammation	[148]
Endostar	CCl_4_-induced liver fibrosis in male BALB/c mice	Improving liver function and reducing liver inflammation	[149]
		Inhibiting a-SMA protein expression	
		Reducing collagen accumulation	
HS-173	CCl_4_-induced liver fibrosis in male BALB/c mice	Blocking the PI3K/Akt pathway	[142]
		Promoting HSC apoptosis	
		Inhibiting the expression of fibrotic mediators	
Ruxolitinib	CCl_4_-induced liver fibrosis in male C57BL/6 mice	Blocking JAK1/2 pathway	[150]
	TAA-induced liver fibrosis in male C57BL/6 mice	Inhibiting the activation of HSCs	
Pantoprazole (PPZ)	BDL-induced liver fibrosis in male SD rats	Promoting the proteasome-dependent degradation and ubiquitination of YAP	[151]
		Inhibiting HSC activation	
		Disrupting the interaction between OTUB2 and YAP	
Tenofovir disoproxil fumarate (TDF)	TAA-induced liver fibrosis in male C57BL/6 mice	Downregulating the PI3K/Akt/mTOR signaling pathway	[96]
		The apoptosis of activated HSCs	
Selonsertib	DMN-induced liver fibrosis in male SD rats	Blocking ASK1/MAPK signaling	[152]
		Inducing apoptosis of HSCs	
Rilpivirine (RPV)	CCl_4_-induced fibrosis in female C57BL/6J mice	Selective STAT1-dependent induction of apoptosis in HSCs	[153]
	BDL-induced fibrosis in female C57BL/6J mice	Promoting liver regeneration	

## Data Availability

Not applicable.

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
