# Peer review of "Liver Fibrosis Resolution: From Molecular Mechanisms to Therapeutic Opportunities"

_ijms, 2023, doi:10.3390/ijms24119671_

Round 1

Reviewer 1 Report

This paper aims to present the current knowledge and understanding of the molecular mechanisms that induces liver fibrosis, and some of the therapeutic approaches that have been suggested to slow down or reverse the fibrotic process. The paper covers the topic from a general point of view, include a number of recent papers in the field, and could be for the interest of readers interested in liver diseases. The novelty of the review resides in that authors have included the antifibrotic effect of natural compounds, more precisely some compounds from the traditional Chinese medicine.

In order to improve the manuscript, as the authors mention in the text, “a significant number of non-coding RNAs (ncRNAs) regulate the development of liver fibrosis”. However, there is little information about siRNA and miRNAs in the text. Nowadays, there are a number of papers or specific review papers that could address this information. Related to this field, another topic that is not covered in the paper is the role of exosomes/extracellular vesicles, that have been implicated in liver fibrosis and may also inhibit the progression of fibrosis. Hepatocytes and non-parenchymal cells, including HSCs, LSEC and cholangiocytes, secrete exosomes and regulate liver remodeling after injury.

Related to the regression of Liver Fibrosis, authors mention that “Antiviral therapy, for instance, exhibited regression of fibrosis in patients with viral hepatitis related liver fibrosis, indicating that the underlying cause of the fibrosis can be treated to reverse liver fibrosis”. It should be described or discussed if it has been published the same effect in the reversion of the disease when the underlying cause of the disease is alcohol, metabolic diseases, biliary diseases, hepatic toxins, autoimmune disease or others. For example, regression of liver fibrosis has not been observed in NASH cirrhotic patients.

In Page 7 lines 306-310, authors mention that “An enzyme called alcohol dehydrogenase (ADH) is mostly expressed in the liver. 95% of the liver's overall activity is accounted for by ADH I, the traditional liver ADH. It was discovered that the expression of ADH I was noticeably higher in aHSCs, and ADH I overexpression might encourage the proliferation, motility, adhesion, and invasion of HSCs”. There are examples as this paragraph in which it is not clear if the example is related to animal models of the disease or human samples. Readers would appreciate that all along the text, authors clearly mention what results come from human clinical studies and which results come from animal models of the disease, as the relevance is different. Afterwards, in the case of the possible treatments, it should be also state if the mentioned results come from human clinical studies or from animal models of the disease.

Specific comments:

-          Page 3 lines 98-99: “MFs are quickly eliminated by apoptosis or inactivation following brief damage”. Authors should consider to remove this sentence from this paragraph.

-          Page 4 lines 150-152. “In addition, HSCs and hepatocytes can interact with LSECs to maintain HSCs quiescence and regenerate hepatocytes, which is essential for fibrosis development. Please, rephrase

-          Page 6 lines 252. “In summary, high levels of ROS are essential for HSCs activation”. Authors should consider to change the term “essential”.

-          Page 6 lines 266-267. “In addition, immune cells, particularly macrophages, went from being pro-inflammatory to a repair state while the local microenvironment did”.  Please, rephrase

-          Page 6 lines 273-275. “Therefore, it is necessary to develop more effective treatments to delay fibrosis progression or regression, which can help prevent the advancement of cirrhosis or cancer”. Please, rephrase.

-          Page 6 lines 283-285. “stimulation of HSCs, which can develop into MFs and trigger significant ECM formation, is a typical feature of liver fibrosis”. This sentence is reiterative, this concept has been previously mentioned in the text.

-          Page 7 lines 300-303.The aberrant proliferation and activation of HSCs is a significant pathogenic cause of liver fibrosis. Hence, liver fibrosis can be prevented or reversed by preventing the proliferation or activation of HSCs. These sentences are reiterative, those concepts have been previously mentioned in the text.

-          Page 7 lines 316, and 321. “Amydrium hainanense (AH) is a plant drug, and its aqueous extract (AHWE) has an antifibrotic effect on mice with liver injury. The primary mechanism through which AHWE had this impact was by preventing HSCs activation and proliferation [79]. Furthermore, a  significant number of non-coding RNAs (ncRNAs) regulate the development of liver fibrosis. For example, Circular RNAs (circRNAs) frequently function as miRNA sponges that control HSCs activation and proliferation and hence participate in the development of liver fibrosis [80]. CircASPH inhibition can inhibit the activation and proliferation of HSCs, while its up-regulation promoted the occurrence of these events [79]”. It looks like those paragraphs are not related to reference 79. Please, confirm the correct references for the mentioned works.  

-          Page 7 lines 338-339. Liver fibrosis treatment relies significantly on endoplasmic reticulum stress (ERS)-mediated apoptosis. Please include a cite for this statement.

-          Page 8 lines 3354-355. Reactivating GATA4 was discovered to change aHSCs into a quiescent phenotype, then reverse liver fibrosis.  Please, rephrase.

-          Page 7 lines 316, and 317. “Furthermore, a significant number of non-coding RNAs (ncRNAs) regulate the development of liver fibrosis”. As relevant, this concept should be mentioned/discussed in the introduction of the liver fibrosis.

-          Page 9 lines 395-396. “In addition, more targets need to be found to provide more options”. This sentence is too vague.

-          Page 9 lines 412-414. Therefore, it is of great significance for liver fibrosis management. Please, rephrase or consider to eliminate this sentence.

-          Page 9 lines 413-414. Numerous cell types and cytokines interact to control the occurrence and  development of liver fibrosis. This sentence is reiterative, this concept has been previously mentioned in the text.

-          Page 15 line 531. The majority of inorganic nanoparticles… “Inorganic” should be “organic”

-          Page 15. Drug Delivery Systems. Authors introduce the concept of nanoparticles,(materials,  types, and so on). However, the majority of the examples that are mentioned of new Drug Delivery Systems applied to liver fibrosis in the text are from liposomes.  

Formal aspects

-          There should be a space between the text and the numbers of the references.  

-          The abbreviation MF is defined in line 76. It should be employed afterwards (line 98, 265, and so on).

-          Page 6 line 273. There is a format problem at the beginning of the sentence.

-          Page 7 lines 338-339. Liver fibrosis treatment relies significantly on endoplasmic reticulum stress (ERS)-mediated apoptosis, And ERS. There should be a “.” instead of a “,”.

-          Page 8. 3.1.3 Inactivation of HSCs . There are format problems in all the paragraphs of point 3.1.3.

-          There is a reiterative use of the term “Therefore”. Please, use a synonym.

-          Page 8 lines 389-390. GNS561 is a novel oral anticancer drug with high hepatotropic properties, And it prevents. There should be a “.” instead of a “,”.

-          Tables 1 and 2. The limits of each row in the tables are not clear.

-          The abbreviation NP is defined in line 515. It should be employed afterwards (line 527,531, and so on).

Author Response

Dear Editor and Reviewers,

Thank you for your useful comments and suggestions on our manuscript. We have modified the manuscript accordingly as follow:

Comment 1: In order to improve the manuscript, as the authors mention in the text, “a significant number of non-coding RNAs (ncRNAs) regulate the development of liver fibrosis”. However, there is little information about siRNA and miRNAs in the text. Nowadays, there are a number of papers or specific review papers that could address this information. Related to this field, another topic that is not covered in the paper is the role of exosomes/extracellular vesicles, that have been implicated in liver fibrosis and may also inhibit the progression of fibrosis. Hepatocytes and non-parenchymal cells, including HSCs, LSEC and cholangiocytes, secrete exosomes and regulate liver remodeling after injury.

Response: Thank you for this very insightful comment. We have added some content about non-coding RNAs (ncRNAs) (line 97-100) and exosomes (line 100-107, line 158-162 etc.) in the process of liver fibrosis to the manuscript.

Comment 2: Related to the regression of Liver Fibrosis, authors mention that “Antiviral therapy, for instance, exhibited regression of fibrosis in patients with viral hepatitis related liver fibrosis, indicating that the underlying cause of the fibrosis can be treated to reverse liver fibrosis”. It should be described or discussed if it has been published the same effect in the reversion of the disease when the underlying cause of the disease is alcohol, metabolic diseases, biliary diseases, hepatic toxins, autoimmune disease or others. For example, regression of liver fibrosis has not been observed in NASH cirrhotic patients.

Response: Thank you for your suggestion. It is of great significance to explore this problem for the regression of liver fibrosis. However, I have not found relevant clinical data reporting regression of human liver fibrosis when the underlying cause is alcohol, metabolic disease, biliary tract disease, liver toxins. So, I have not discussed that in this manuscript.

Comment 3: In Page 7 lines 306-310, authors mention that “An enzyme called alcohol dehydrogenase (ADH) is mostly expressed in the liver. 95% of the liver's overall activity is accounted for by ADH I, the traditional liver ADH. It was discovered that the expression of ADH I was noticeably higher in aHSCs, and ADH I overexpression might encourage the proliferation, motility, adhesion, and invasion of HSCs”. There are examples as this paragraph in which it is not clear if the example is related to animal models of the disease or human samples. Readers would appreciate that all along the text, authors clearly mention what results come from human clinical studies and which results come from animal models of the disease, as the relevance is different. Afterwards, in the case of the possible treatments, it should be also state if the mentioned results come from human clinical studies or from animal models of the disease.

Response: Thank you very much for your correction. The content of this section has been rewritten, and the full text has been checked for similar errors to indicate that the experimental results are derived from clinical data or animal model studies.

  1. Specific comments:

-         1. Page 3 lines 98-99: “MFs are quickly eliminated by apoptosis or inactivation following brief damage”. Authors should consider to remove this sentence from this paragraph.

Response: Thank you for your careful check and correction. We have removed this sentence form the manuscript.

-         2. Page 4 lines 150-152. “In addition, HSCs and hepatocytes can interact with LSECs to maintain HSCs quiescence and regenerate hepatocytes, which is essential for fibrosis development. Please, rephrase

Response: We gratefully appreciate for your valuable suggestion. We have rephrased this sentence in the manuscript as follows: In addition, the interaction among HSCs, hepatocytes and LSECs could maintain HSCs quiescence and regenerate hepatocytes, which is essential for fibrosis development. 

-         3. Page 6 lines 252. “In summary, high levels of ROS are essential for HSCs activation”. Authors should consider to change the term “essential”.

Response: We gratefully appreciate for your valuable suggestion. We have changed this word into “important” in the manuscript.

-          4. Page 6 lines 266-267. “In addition, immune cells, particularly macrophages, went from being pro-inflammatory to a repair state while the local microenvironment did”.  Please, rephrase

Response: We gratefully appreciate for your valuable suggestion. We have rephrased this sentence in the manuscript as follows: When the stimulation of liver injury is removed, the liver can activate repair mechanisms that inhibit MFs activation and change immune cells, particularly macrophages, from being pro-inflammatory to a repair state.

-         5. Page 6 lines 273-275. “Therefore, it is necessary to develop more effective treatments to delay fibrosis progression or regression, which can help prevent the advancement of cirrhosis or cancer”. Please, rephrase.

Response: We gratefully appreciate for your valuable suggestion. We have rephrased this sentence in the manuscript as follows: Consequently, it is necessary to develop more effective therapies to delay fibrosis progression or promote regression of liver fibrosis, which can help prevent the advancement of cirrhosis or cancer.

-          6. Page 6 lines 283-285. “stimulation of HSCs, which can develop into MFs and trigger significant ECM formation, is a typical feature of liver fibrosis”. This sentence is reiterative, this concept has been previously mentioned in the text.

Response: We gratefully appreciate for your valuable suggestion. We have removed this sentence form the manuscript.

-          7. Page 7 lines 300-303.The aberrant proliferation and activation of HSCs is a significant pathogenic cause of liver fibrosis. Hence, liver fibrosis can be prevented or reversed by preventing the proliferation or activation of HSCs. These sentences are reiterative, those concepts have been previously mentioned in the text.

Response: We gratefully appreciate for your valuable suggestion. We have removed this sentence form the manuscript.

-          8. Page 7 lines 316, and 321. “Amydrium hainanense (AH) is a plant drug, and its aqueous extract (AHWE) has an antifibrotic effect on mice with liver injury. The primary mechanism through which AHWE had this impact was by preventing HSCs activation and proliferation [79]. Furthermore, a  significant number of non-coding RNAs (ncRNAs) regulate the development of liver fibrosis. For example, Circular RNAs (circRNAs) frequently function as miRNA sponges that control HSCs activation and proliferation and hence participate in the development of liver fibrosis [80]. CircASPH inhibition can inhibit the activation and proliferation of HSCs, while its up-regulation promoted the occurrence of these events [79]”. It looks like those paragraphs are not related to reference 79. Please, confirm the correct references for the mentioned works.  

Response: Thank you for your careful check and correction. The reference has been replaced with the correct one. Meanwhile, the references of the full text have been checked and corrected.

-          9. Page 7 lines 338-339. Liver fibrosis treatment relies significantly on endoplasmic reticulum stress (ERS)-mediated apoptosis. Please include a cite for this statement.

Response: Thank you for your careful check and correction. The reference has been added in the manuscript.

-          10. Page 8 lines 3354-355. Reactivating GATA4 was discovered to change aHSCs into a quiescent phenotype, then reverse liver fibrosis.  Please, rephrase.

Response: We gratefully appreciate for your valuable suggestion. We have rephrased this sentence in the manuscript as follows: Research has discovered that reactivation of GATA4 couild change aHSCs into a quiescent phenotype, and then reverse liver fibrosis.

-          11. Page 7 lines 316, and 317. “Furthermore, a significant number of non-coding RNAs (ncRNAs) regulate the development of liver fibrosis”. As relevant, this concept should be mentioned/discussed in the introduction of the liver fibrosis.

Response: We gratefully appreciate for your valuable suggestion. We have rewritten this section to the introduction of the liver fibrosis in the manuscript.

-          12. Page 9 lines 395-396. “In addition, more targets need to be found to provide more options”. This sentence is too vague.

Response: We gratefully appreciate for your valuable suggestion. We have rewritten this sentence in the manuscript as follows: In addition, more targets need to be found to provide more options for liver fibrosis reversing.

-          13. Page 9 lines 412-414. Therefore, it is of great significance for liver fibrosis management. Please, rephrase or consider to eliminate this sentence.

Response: We gratefully appreciate for your valuable suggestion. We have removed this sentence form the manuscript.

-          14. Page 9 lines 413-414. Numerous cell types and cytokines interact to control the occurrence and  development of liver fibrosis. This sentence is reiterative, this concept has been previously mentioned in the text.

Response: We gratefully appreciate for your valuable suggestion. We have removed this sentence form the manuscript.

-          15. Page 15 line 531. The majority of inorganic nanoparticles… “Inorganic” should be “organic”

Response: Thank you for your careful check and correction. We have changed this word into “organic” in the manuscript.

-          16. Page 15. Drug Delivery Systems. Authors introduce the concept of nanoparticles,(materials,  types, and so on). However, the majority of the examples that are mentioned of new Drug Delivery Systems applied to liver fibrosis in the text are from liposomes.  

Response: Thank you for this very insightful comment. We have added the corresponding examples of other delivery systems in the manuscript.

  1. Formal aspects:

-          1. There should be a space between the text and the numbers of the references.  

Response: Thank you for your careful check and correction. We have added a space between the text and the numbers of the references of the full text.

-          2. The abbreviation MF is defined in line 76. It should be employed afterwards (line 98, 265, and so on).

Response: Thank you for your careful check and correction. We have corrected this abbreviation. And similar errors in the full text have been checked and corrected.

-          3. Page 6 line 273. There is a format problem at the beginning of the sentence.

Response: Thank you for your careful check and correction. We have corrected this format problem in the manuscript. And similar errors in the full text have been checked and corrected.

-          4. Page 7 lines 338-339. Liver fibrosis treatment relies significantly on endoplasmic reticulum stress (ERS)-mediated apoptosis, And ERS. There should be a “.” instead of a “,”.

Response: Thank you for your correction. We have corrected this problem in the manuscript.

-          5. Page 8. 3.1.3 Inactivation of HSCs . There are format problems in all the paragraphs of point 3.1.3.

Response: Thank you for your careful check and correction. We have corrected these format problems in all the paragraphs of point 3.1.3. And similar errors in the full text have been checked and corrected.

-          6. There is a reiterative use of the term “Therefore”. Please, use a synonym.

Response: We gratefully appreciate for your valuable suggestion. We have replaced the use of “Therefore” with “Hence”, “Thus”, “Accordingly” etc. See the manuscript for details, please.

-          7. Page 8 lines 389-390. GNS561 is a novel oral anticancer drug with high hepatotropic properties, and it prevents. There should be a “.” instead of a “,”.

Response: Thank you for your correction. We have corrected this error in the manuscript.

-         8.  Tables 1 and 2. The limits of each row in the tables are not clear.

Response: Thank you for your careful check and correction. We have adjusted the tables to make the limits of each row clearer. See the manuscript for details, please.

-          9. The abbreviation NP is defined in line 515. It should be employed afterwards (line 527,531, and so on).

Response: Thank you for your careful check and correction. We have corrected this abbreviation. And similar errors in the full text have been checked and corrected.

Finally, we thank you once again for your valuable comments and suggestions. We deeply appreciate your consideration of our manuscript. If you have any queries, please do not hesitate to contact us.

Thank you and best regards.

Sincerely yours,

Liling Tang

PhD, Professor

Reviewer 2 Report

I have reviewed the manuscript Liver Fibrosis Resolution: From Molecular Mechanisms to Therapeutic Opportunities by Qiying Pei, Qian Yi, and Liling Tang.

This Review is a comprehensive review in which the most robust observation is to prevent chronic liver disease to avoid fibrosis, but if this is already present, then the option is the resolution of fibrosis by inhibiting the HSC activation or promoting their apoptosis, thus reversing the accumulation of extracellular matrix; also, the authors compile the current molecular routes and targets to regulate such profibrogenic events in the liver with novel therapeutic drugs.

I think the manuscript is a valuable work that can be improved; some suggestions are commented to the authors.

a)    Please correct some typing mistakes, such as:… liver. 95%, this is on line 307, third paragraph, and page 7. Also, CCl4 is incorrect; the number 4 must be written subscript as CCl4 since this is the number of chloride atoms in the molecule, as the IUPAQ rules demand. Moreover, correct Ca2+-dependent while it is Ca2+ with oxidation number in superscript; this is on page 7, line 340, fifth paragraph. Besides, amend please some changes in letter size throughout the manuscript, as well as some unnecessary uppercase and lowercase use in some words.

b)    What is the chemical content of Amydrium hainanense? Please spell circASPH out ultimately, derived from the aspartate β-hydroxylase (ASPH) gene locus. Also, the authors must be sure that all the scientific names of plants shall be written in superscript initials for genus and lower script initials for specie and italicized or underlined since it is Latin, not English, such as Saussurea lappa and Angelica sinensis.

c)    The authors should describe the scientific names of the plants or other constituents and the chemical compounds of the Chinese formulas to know them; furthermore, to avoid the illegal use of plants or animal species in extinction danger by not promoting their medicinal use despite their therapeutic efficacy.

d)    Add a figure with two panels, one depicting the leading natural products (secondary metabolites) and the second showing the synthetic or functionalized nanoparticles described in the text, which may be more attractive and didactic for comparison purposes to interested readers.

Some typing mistakes must be corrected to improve this journal's English grammar.

Author Response

Dear Editor and Reviewers,

Thank you for your useful comments and suggestions on our manuscript. We have modified the manuscript accordingly as follow:

Comment 1: Please correct some typing mistakes, such as:… liver. 95%, this is on line 307, third paragraph, and page 7. Also, CCl4 is incorrect; the number 4 must be written subscript as CCl4 since this is the number of chloride atoms in the molecule, as the IUPAQ rules demand. Moreover, correct Ca2+-dependent while it is Ca2+ with oxidation number in superscript; this is on page 7, line 340, fifth paragraph. Besides, amend please some changes in letter size throughout the manuscript, as well as some unnecessary uppercase and lowercase use in some words.

Response: We gratefully appreciate for your correction. We have corrected CCl4 and Ca2+ in the manuscript. And we have checked the full text and corrected the use of superscripts, subscripts and words in upper and lower case.

Comment 2:  What is the chemical content of Amydrium hainanense? Please spell circASPH out ultimately, derived from the aspartate β-hydroxylase (ASPH) gene locus. Also, the authors must be sure that all the scientific names of plants shall be written in superscript initials for genus and lower script initials for specie and italicized or underlined since it is Latin, not English, such as Saussurea lappa and Angelica sinensis.

Response: We gratefully appreciate for your meticulous check. Amydrium hainanense is a botanical drug, which is popularly known as China’s unique climbing Fujimoto. And its aqueous extract has an antifibrotic effect on mice with liver injury. But I have not found relevant literature that reports the chemical content of Amydrium hainanense or its aqueous extract. CircASPH has been spelled out ultimately in the manuscript. All the scientific names of plants have been written italicized. And we have checked the full text and corrected similar errors.

Comment 3: The authors should describe the scientific names of the plants or other constituents and the chemical compounds of the Chinese formulas to know them; furthermore, to avoid the illegal use of plants or animal species in extinction danger by not promoting their medicinal use despite their therapeutic efficacy.

Response: We gratefully appreciate for your valuable suggestion. Relevant information, such as the composition of FZHY and the chemical composition of JY5, has been added in the manuscript.

Comment 4: Add a figure with two panels, one depicting the leading natural products (secondary metabolites) and the second showing the synthetic or functionalized nanoparticles described in the text, which may be more attractive and didactic for comparison purposes to interested readers.

Response: We gratefully appreciate for your valuable suggestion. According to the content of the manuscript, we have added the following pictures to summarize the regression of liver fibrosis and the therapeutic compounds.

Finally, we thank you once again for your valuable comments and suggestions. We deeply appreciate your consideration of our manuscript. If you have any queries, please do not hesitate to contact us.

Thank you and best regards.

Sincerely yours,

Liling Tang

PhD, Professor

Round 2

Reviewer 1 Report

The authors have properly answered to the questions and suggestions formulated by this referee. The revised version of the manuscript has been improved. I would like to thank the authors for the effort.

As a final comment, as the authors comment, they “have not found relevant clinical data reporting regression of human liver fibrosis when the underlying cause is alcohol, metabolic disease, biliary tract disease, liver toxins. For this referee, the incorporation of this concept in the final version of the manuscript would improve the paper in order to be clear with the potential readers of the paper.